# Exploration of Breakdown Strength Decrease and Mitigation of Ultrathin Polypropylene

**DOI:** 10.3390/polym15102257

**Published:** 2023-05-10

**Authors:** Daniel Q. Tan, Yichen Liu, Xiaotian Lin, Enling Huang, Xi Lin, Xudong Wu, Jintao Lin, Ronghai Luo, Tianxiang Wang

**Affiliations:** 1Department of Materials Science and Engineering, Guangdong Technion-Israel Institute of Technology, 241 Daxue Road, Shantou 515063, Chinaxi.lin@gtiit.edu.cn (X.L.);; 2Guangdong Provincial Key Laboratory of Materials and Technologies for Energy Conversion, Guangdong Technion-Israel Institute of Technology, 241 Daxue Road, Shantou 515063, China; 3Department of Materials Science and Engineering, Technion-Israel Institute of Technology, Haifa 3200003, Israel; 4Xiamen Faratronic Ltd., Co., Xiamen 361028, China; ljt0101@sina.com (J.L.);

**Keywords:** polymer, breakdown strength, film thickness, film stretching, capacitor

## Abstract

Polypropylene film is the most important organic dielectric in capacitor technology; however, applications such as power electronic devices require more miniaturized capacitors and thinner dielectric films. The commercial biaxially oriented polypropylene film is losing the advantage of its high breakdown strength as it becomes thinner. This work carefully studies the breakdown strength of the film between 1 and 5 microns. The breakdown strength drops rapidly and hardly ensures that the capacitor reaches a volumetric energy density of 2 J/cm^3^. Differential scanning calorimetry, X-ray, and SEM analyses showed that this phenomenon has nothing to do with the crystallographic orientation and crystallinity of the film but is closely related to the non-uniform fibers and many voids produced by overstretching the film. Measures must be taken to avoid their premature breakdown due to high local electric fields. An improvement below 5 microns will maintain a high energy density and the important application of polypropylene films in capacitors. Without destroying the physical properties of commercial films, this work employs the ALD oxide coating scheme to augment the dielectric strength of a BOPP in the thickness range below 5 μm, especially its high temperature performance. Therefore, the problem of the reduction in dielectric strength and energy density caused by BOPP thinning can be alleviated.

## 1. Introduction

Polypropylene film is currently the most valued organic dielectric material in capacitor technology. Its advantages of having a low dielectric loss, high breakdown strength, ultra-thin thickness, and relatively low cost make it dominant in high-end technology applications such as new energy, high-speed railway transportation, high-voltage transmission, electric vehicles, oil exploration, aerospace, and electromagnetic weapons. Produced by stretching, biaxially oriented polypropylene film (BOPP) has gradually replaced paper dielectrics since the 1960s; after the 1980s, it became the main dielectric film for film/aluminum foil and metallized film capacitors [1,2]. Early studies found that the breakdown strength of BOPP increased as the thickness decreased in the range of 8–80 μm (μm) [3]. However, recent studies have found that the breakdown strength of BOPP reaches its highest value in the range of 5–15 μm, and the specific results depend on the test method and film manufacturing process [4,5]. The ever-increasing need in miniaturized electronic devices requires thinner dielectric films with high dielectric strengths for smaller volumes in capacitor modules [6,7,8]. Polymer dielectrics, with the advantages of easy scaleup processing, a light weight, low cost, high dielectric strength, and good flexibility, offer a broad range of capacitance, a high voltage rating, graceful failure reliability, and self-clearing potential for capacitor manufacturing. For compact capacitors, it is critical to ensure that the film is as thin as possible and its dielectric strength as high as possible. Since the volumetric capacitance and therefore the volumetric energy density are inversely proportional to the dielectric film thickness, reducing the dielectric thickness is greatly beneficial. However, a further reduction in film thickness leads to a decrease in dielectric strength [9,10,11]. The authors have been concerned about this thickness-dependent dielectric phenomenon because the decrease in breakdown strength due to thinning will become a huge challenge for capacitor miniaturization and high-voltage design [12]. The common breakdown mechanisms in polymeric materials include electron avalanche, thermal runaway, electromechanical failure, and long-term ionic migration. Polymer films containing various molecular chains, contaminants, interfaces, and surfaces can break down due to complicated factors. Unfortunately, most individuals who pay attention to thickness factors are capacitor design engineers.

Various attempts toward the development of high-temperature polymers such as polytetrafluoro-ethylene (PTFE), polyetherimide (PEI), polyethylene naphthalate (PEN), fluorene polyester (FPE), and polyimide (PI) exist. Some of these polymers possess a lower dielectric strength, higher dielectric loss, or lack a self-clearing capability in spite of commercialization. Some high-temperature dielectrics are still being investigated in research laboratories despite holding promise for more exceptional dielectric properties and high-temperature ratings. Their challenges, such as their dielectric strength, dielectric loss, and high-quality film processing, remain [13,14,15,16,17]. In addition, some researchers have attempted to increase the energy density of BOPP by loading high-permittivity particles onto the outer surface of BOPP film via plasma-assisted solution coating, magnetron sputtering, or corona treatment [18,19,20]. However, the temperature of BOPP remains below 125 °C. Given the slow progress in research on high-temperature polymer dielectric films and the commercial manufacturing of films, it is necessary to strengthen the in-depth research on ultra-thin polypropylene films. Currently, the production process of capacitor-grade polypropylene films has not only been upgraded on a large scale but has also greatly reduced the thickness of BOPP. Foreign and domestic researchers have obtained ultra-thin test materials of 1.6 and 1.9 μm, respectively, and capacitors have been manufactured using 2.4 μm BOPP [21,22]. Figure 1a shows that both domestic and foreign films (copper-clad laminate, Farad Electronic, Bollore) exhibit a decrease in breakdown strength during the thinning process [23]. However, there are no reports on how reducing BOPP to below 2.4 μm affects their breakdown strength, and the detailed relationship and reasons for the decrease in the breakdown strength of ultra-thin BOPP in the range of 1–5 μm are not clear. Whether the decrease in the breakdown strength of ultra-thin BOPP can be avoided also needs further study. In this work, the author uses commercial BOPP film and further reduces the film thickness through stretching to study the behavior of the breakdown strength with decreasing thickness and explores the reasons for the decrease through microscopic characterization in preparation for improving the breakdown strength of ultra-thin BOPP. Atomic-layer=deposition technology is used to coat the oxide nano-layer, which mitigates the reduction in the breakdown strength of the ultra-thin BOPP.

## 2. Experimental Section

This article tests polypropylene films (BOPP and HCPP) of different thicknesses provided by Xiamen Faratronic Co., Ltd. of China and Bollore of the United States, respectively. The film provided by Faratronic was manufactured by Quanzhou Jia Deli Electronic Materials Co., Ltd. and Toray, Japan, and has an average molecular weight of 60–70 kg/mol. The films were further thinned using a laboratory-prepared heating plate, which was stretched along the MD direction at a constant temperature (150–160 °C). ALD coating was carried out using a Benchtop GESTar XT-D ALD system commercially provided by Arradiance, Inc. Aluminum oxide (Al_2_O_3_) from Nanjing Ai Mou Yuan Scientific Equipment Co. Lt. (Nanjing, China), and high-performance liquid-chromatography-grade H_2_O precursors were targeted at various depositing conditions.

Film images were captured using a Zeiss Sigma-500 field emission scanning electron microscope (FESEM) from Germany. The direct current breakdown behavior was tested using a PK-CPE1801 ferroelectric polarization ring and a dielectric breakdown test system (PolyK, State College, CA, USA) equipped with a Trek high-voltage amplifier (610E), measuring the sample immersed in a constant-temperature oil bath system with a voltage rise rate of 500 V/s, using a metal ball-plan electrode structure with a ball diameter of 6 mm. Differential scanning calorimetry (DSC) measurements were performed using a TA Instruments Q200 DSC in the 20 °C to 200 °C temperature range, using a rate of 5 °C/min.

## 3. Results and Discussion

### 3.1. Electrical Breakdown Behavior of Ultra-Thin BOPP

This study presents a detailed investigation into the electrical breakdown issues of BOPP films stretched to below 4 μm. In order to obtain the breakdown strength of BOPP films below 4 μm, 3.8 μm, and 2.4 μm thick, films were gradually stretched to their lowest achievable thickness, such as from 3.8 μm to 2.4 μm and from 2.4 μm to 1.3 μm. Figure 1b shows the Weber distribution of the original film’s breakdown strength between 5.8 μm and 2.4 μm and its decrease with thickness. It can be observed that the gradual thinning of BOPP films causes a gradual decrease in their breakdown strength. When the film thickness is reduced to below 2.4 μm, the breakdown strength sharply decreases, with a 37% reduction in breakdown strength for a 1.3 μm thick film compared to thicker films. According to this trend, the breakdown strength will decrease to below 250 kV/mm after 0.5 μm, which is close to the electric field strength required to eliminate weak points in the film. This would result in a loss of dielectric advantages, making it difficult to use for capacitor fabrication.

The reduction in dielectric strength with an increase in the film thickness is generally attributed to the contaminants in thick films. However, a thinning operation would decrease the arial density of the contaminants by stretching the polymers. The lower strength cannot be fully explained by simply considering contaminants since the Weibull analysis on the data from the small electrode was used. It is yet to be undersood why the strength is smaller when the thickness is very thin.

According to the formula for volumetric energy density (1/2ε_0_εV^2^/d^2^, where ε is the dielectric constant, V is the voltage, and d is the film thickness), the energy density increases significantly as the thickness decreases. Theoretically, BOPP is expected to achieve an energy density of over 5 J/cm^3^ (Figure 2). However, the breakdown strength of the film is limited at the working voltage, making it impossible to support higher theoretical energy densities. For a 2 µm thick BOPP film, the maximum breakdown strength tested was 630 kV/µm, but factors such as increasing the electrode area, process flow, and packaging density led to a decrease in the rated voltage, resulting in a decrease in the breakdown strength and energy density of the capacitor to 375 kV/µm and 1.3 J/cm^3^, respectively, or even lower (calculated based on a 60% reduction). With further thinning of the film (0.5–2 µm), the energy density of the capacitor can be expected to greatly decrease to less than 1 J/cm^3^ due to the significant reduction in breakdown strength (<400 kV/µm). Additionally, BOPP films are limited to low temperatures, and increasing the temperature would significantly reduce their breakdown strength [24]. Due to its low breakdown strength, ultra-thin BOPP films will lose their advantages in applications for capacitors.

To understand the underlying reasons for this phenomenon, XRD and SEM analyses were used to characterize the semi-crystalline BOPP film and investigate the causes of the decrease in breakdown strength due to thinning. The tests showed that the BOPP films provided by Far Eastern New Century Corporation maintained a consistent crystal orientation regardless of thickness (Figure 3). The (040) diffraction peak located at 17 degrees exhibited the highest intensity and maintained a similar height and shape compared to the (110) peak during the thinning process (1.3 µm curve). This suggests that the stretching process did not disrupt the original biaxial crystal orientation of the BOPP. In addition, the thinning of the BOPP films did not lead to the appearance of the β-phase, indicating that the decrease in breakdown strength was not related to a phase transition [25].

As is well known, defects such as contaminants in thick polymer films (over ten microns) play an important role in reducing breakdown strength. However, surface morphology and defects become more important as the film thickness decreases to a few microns. It is necessary to use SEM imaging technology to directly display the surface morphology and microstructure of thin film materials. Figure 4 shows that 2.4 µm thick BOPP undergoes changes as it is stretched thinner. Before stretching, the surface of the film exhibits a rough morphology and curved stripes generated during the film formation process (Figure 4a), with local areas showing uneven surfaces and fibers of varying thicknesses (Figure 4b). After the thickness is reduced to below 2.4 µm, the surface exhibits uneven, ridge-like fiber structures and non-dense regions. Figure 4c,d show nanometer-scale gaps and nanopores [26]. These defect features are likely the main reason for the sharp decrease in breakdown strength below 2.4 µm, as the breakdown strength of air is much lower than that of polymers, making it easy for ions to be ionized and discharged under high pressure. In fact, the breakdown strength of porous membranes is about 100 kV/mm or less (Figure 1). The presence of and increase in micropores directly reduces the breakdown strength of thin films. Suppressing micropores can help improve the dielectric breakdown strength of BOPP. Before high-temperature films become available, once BOPP films no longer have the advantage of a high breakdown strength, the goal of miniaturizing capacitors will be difficult to achieve.

Another mechanism of breakdown to be discussed is the crystallinity and crystalline orientation in the stretched BOPP films. DSC testing was carried out to determine the crystallinity in the 2.4 µm and 4.8 µm BOPP films (no figures included). The first heating endothermic melting peaks occur at ~172.7 (onset at 162.7 °C) and ~169.1 °C (onset at 154 °C), with the enthalpies of 98.36 and 123.6 J/g, respectively. Their exothermic crystallization peaks occur at 124.3 and 122.8 °C with enthalpies of 112.4 and 132.3 J/g, respectively. According to the enthalpies of both crystallization and melting of BOPP, the crystallinity for both films were calculated to be similarly high (47–48%). It is well known that the commercial BOPP films for capacitor manufacturing generally exhibit a crystallinity of >45% [27,28]. Stretching films further, to below 2 µm at temperatures of 150–160 °C, would not change the crystallinity. Therefore, crystallinity may not be the major factor in the observed reduction in the breakdown strength. Defect generation and crystalline orientation are more concerns that affectthe breakdown strength. Reducing the rated voltage of capacitors will increase the number of capacitors in series and parallel and the volume of the module. Therefore, the academic and industrial communities need to work together actively to study the role of BOPP surface morphology and defects in depth and propose innovative solutions as soon as possible to prevent the decrease in breakdown strength with film thickness.

Experts have reported that the surface coating of inorganic nanoparticles can improve the breakdown strength of BOPP to a certain extent [29,30,31]. On one hand, the high dielectric constant of the oxide reduces the localized electric field strength on the surface of BOPP, thereby reducing the degree of surface charge accumulation. On the other hand, the high elastic modulus of the oxide directly contributes to the high breakdown strength. However, more in-depth and detailed research is needed to understand the high-temperature dielectric properties of modified BOPP. It will be worthwhile to systematically investigate whether the molecular chains of the low-temperature phases (amorphous and crystalline) in BOPP lose their mobility due to the clamping effect of the high modulus coating, which helps to stabilize the high-temperature dielectric properties.

### 3.2. Mitigation Scheme of Breakdown Strength

In order to suppress the surface defects and roughness in the biaxially stretched BOPP film, thus maintaining or improving the breakdown strength of the film, this work uses the method of nano-coating to modify the surface of the BOPP. Figure 5a shows the DC breakdown strength of BOPP films with different thicknesses before and after depositing nanometer-thick alumina on both sides. The test at room temperature shows that tens-of-nanometers-thick ALD alumina can significantly improve the ultra-thin BOPP, and the breakdown strength can be increased by more than 10%. This improvement appears to be more pronounced in the high temperature range. Figure 5b shows that the 3.8 μm BOPP film can maintain its high breakdown strength with an ALD alumina coating tens of nanometers thick between 105 and 140 °C. At room temperature and a high temperature of 140 °C, the breakdown strength of the modified film is 837.7 and 650.4 kV/mm, respectively. Compared with the PP matrix, the breakdown strength is increased by 11% and 22%, respectively. The scanning electron micrographs in Figure 4 show that the alumina deposited on the surface of BOPP tightly packs together to form a dense film, covering the void defects on the surface of BOPP. It is worth noting that different surface modification approaches may be positive as well. This includes the UV treatment of BOPP film, which enhanced the breakdown strength by consuming the antioxidant [32]. Surface oxide coating using a solution or the magnetron sputtering method also showed a positive effect despite its limited influence on the operation temperature of BOPP [18,19].

The reason for the increase in breakdown strength was explored in the author’s previous reports. On one hand, it was believed that the high dielectric constant of alumina reduces the local electric field intensity on the surface of BOPP and reduces the accumulation of surface charges. On the other hand, it was also pointed out that the high elastic modulus of alumina directly contributes to the high breakdown strength [3,4]. In this paper, an XRD diffraction spectrum was used to further characterize the change in the crystallographic orientation of the BOPP film during in situ heating and cooling. Figure 6 shows the difference in the X-ray diffraction spectra of 4.8 µm BOPP before and after ALD alumina deposition. The α crystal phase, represented by the (110), (040), (130), and (111) peaks of the pristine BOPP film, dominate the biaxially stretched crystal orientation. After in situ heating to 115 °C for half an hour, the peak height decreased dramatically, and after half an hour from 150 °C to room temperature, the diffraction peaks no longer returned to their height before heating. This irreversible height change shows that most of the crystal orientation produced by biaxial stretching has been lost, which is a true reflection of the mobility of the molecular chains inside the polypropylene film. In contrast, the BOPP film coated with ALD alumina exhibited a stable diffraction peak, indicating that the α-phase-dominated crystal orientation was stabilized due to the clamping of the alumina thin layer. The specific reason for this may be that the molecular chains of the low-temperature amorphous phase and crystalline phase in BOPP cannot move freely at a high temperature due to the clamping of the high-elastic-modulus alumina, so the α phase is not affected and still maintains its orientation at a high temperature.

In terms of DSC measurement, the onset temperature of the melting curve for ALD-coated BOPP occurs at 158.9 ℃ higher than that of primitive BOPP (153.1 °C). The enthalpy in the ALD-coated BOPP is also less. These results may imply a stable crystalline phase up to higher temperatures, supporting the observed stable XRD peaks at higher temperatures.

The complex dielectric responses of both types of films were also measured as a function of temperature, using a Novocontrol Dielectric Spectrometer. No significant differences were observed except for a slight decrease in the dielectric constant when increasing the temperature from 25 to 140 °C (2.21 to 2.07). Additionally, the dielectric loss remained below 0.001 in the frequency range of measurement.

## 4. Conclusions

Commercial BOPP films face the challenge of a reduction in breakdown strength when their thickness is less than 5 µm. This study was the first to discover that the breakdown strength rapidly decreases when the thin film is stretched to less than 2.4 µm. It is expected that a 1 µm BOPP film will decrease to below 400 kV/mm by further stretching it below melting temperatures, losing its advantage of high dielectric strength, and the capacitor it forms will be unable to provide a volumetric energy density greater than 1 J/cm^3^. The analysis of DSC results suggests no significant difference in crystallinity between the thick and thinner films. XRD demonstrated that the high crystalline orientation of BOPP remains critical for the maintenance of its dielectric strength and operation temperature. SEM characterization suggests that the abnormal morphology of BOPP fibers and the generation of many voids in the films are responsible for the decrease in dielectric strength. It is essential to fully recognize the severity of the loss of the high breakdown strength with further thinning of the BOPP. Utilizing an ALD oxide-coating scheme can improve the dielectric strength of BOPP in the thickness range below 5 μm, especially the high-temperature performance. Therefore, the problems of dielectric strength reduction and energy density reduction caused by BOPP thinning can be mitigated by the ALD oxide coating. This method may extend the utilization of ultra-thin BOPP films below 2 μm in the manufacture of high-performance miniaturized capacitors, and it may also serve as an effective methodology for other dielectric polymers of a lower dielectric strength and glass transition temperatures.

## Figures and Tables

**Figure 1 polymers-15-02257-f001:**
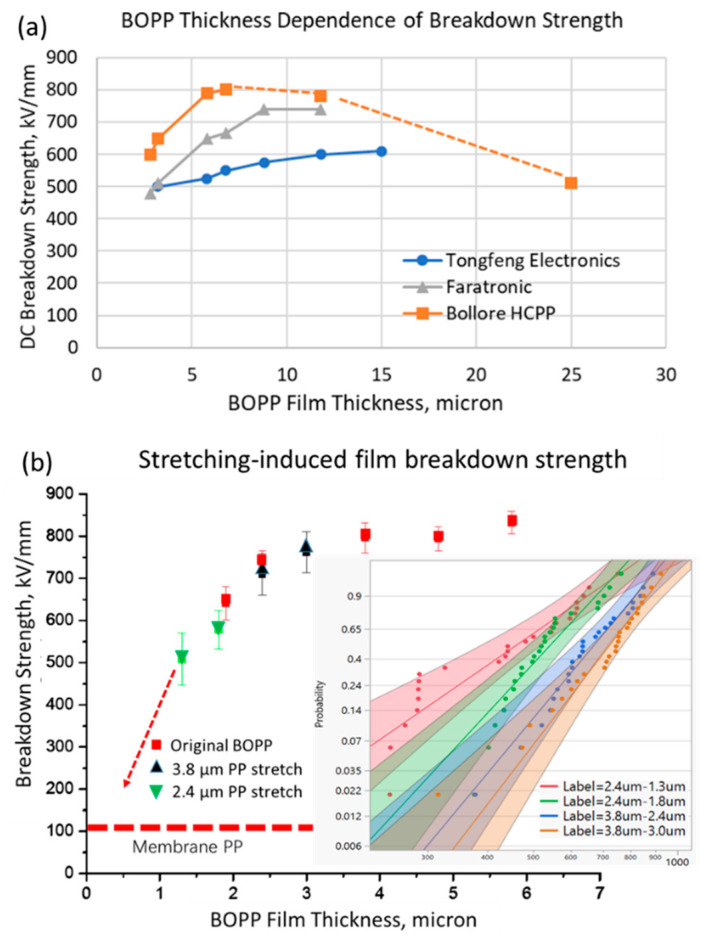
(**a**) DC breakdown strength of different BOPP films as a function of thickness; (**b**) stretched BOPP films in comparison with commercial BOPP films (Weibull distribution in the inset, shape factor β between 5 and 11). Data for domestic films acquired using the standard dielectric strength-testing method with a copper rod of 25 mm in diameter are average values.

**Figure 2 polymers-15-02257-f002:**
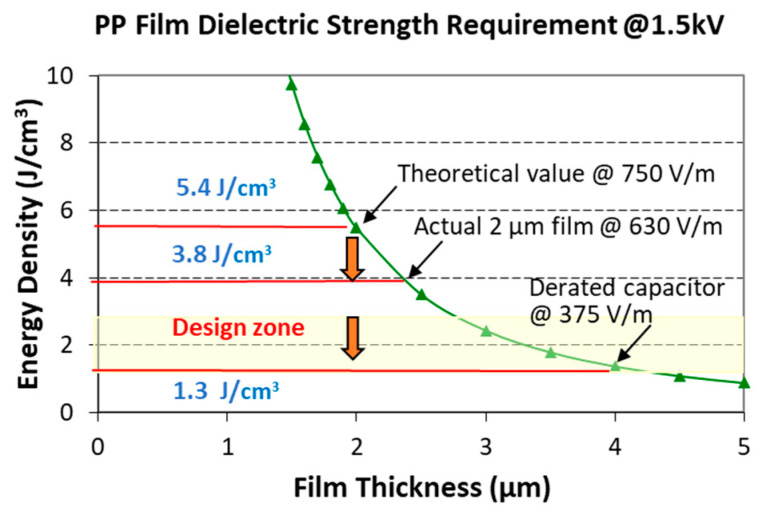
Volumetric energy density of capacitors as a function of film thickness under a designed voltage of 1.5 kV.

**Figure 3 polymers-15-02257-f003:**
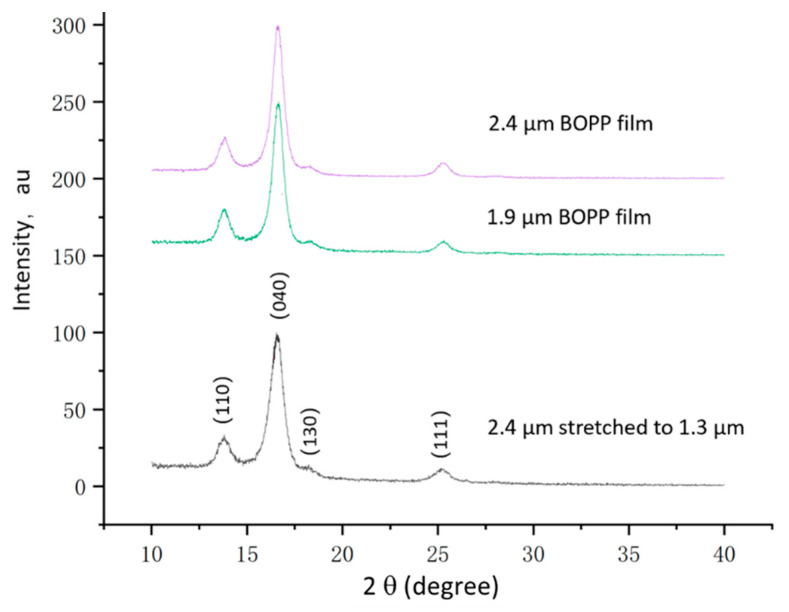
XRD patterns of 2.4, 1.9, and 1.3 µm BOPP films in terms of diffraction angle (2 theta). Four major peaks of BOPP appear below 30 degrees.

**Figure 4 polymers-15-02257-f004:**
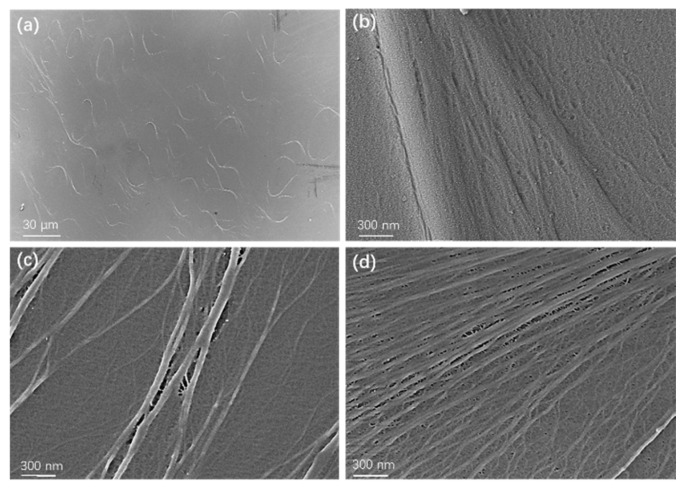
SEM Images of BOPP films before and after stretching: (**a**,**b**) 2.4 µm at two magnifications, (**c**) 1.9 µm and (**d**) 1.3 µm.

**Figure 5 polymers-15-02257-f005:**
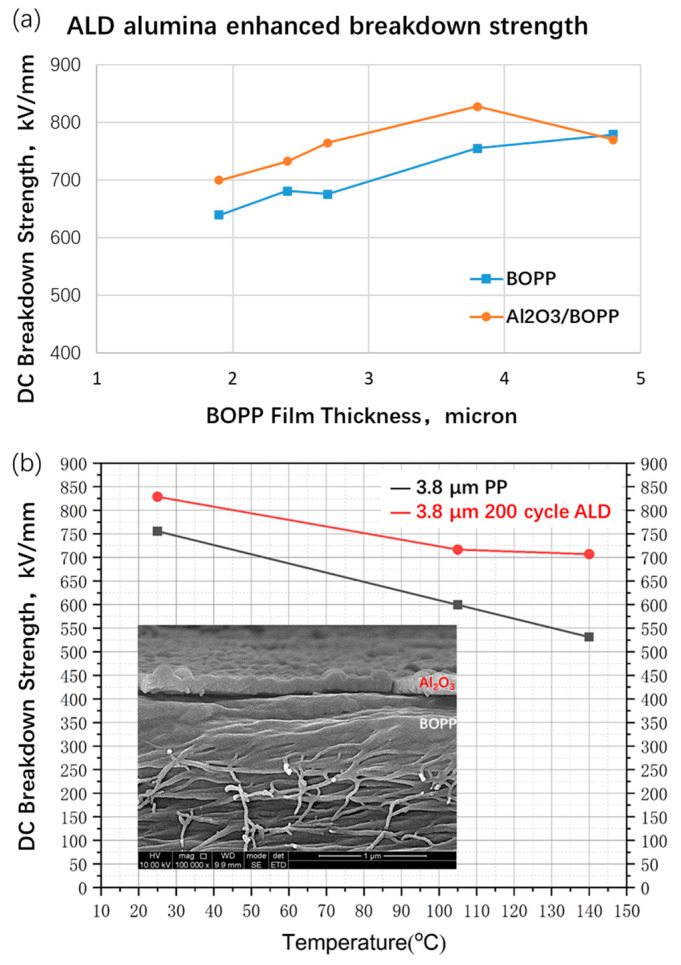
(**a**) Effect of ALD alumina coating on breakdown strength of BOPP films of various thicknesses; (**b**) breakdown strength enhancement at higher temperatures and SEM images of cross section of coated 3.8 µm BOPP.

**Figure 6 polymers-15-02257-f006:**
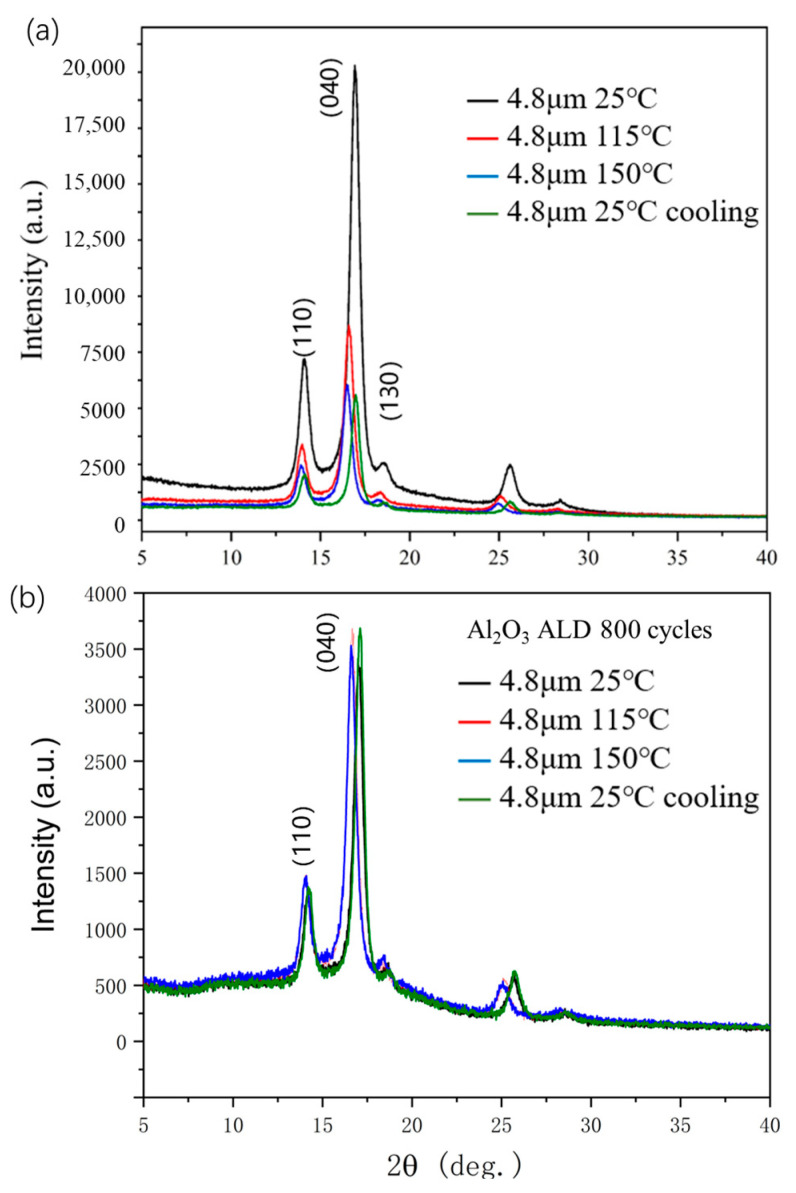
Effect of in situ heat treatment (115, 150 °C for 30 min) on XRD patterns of BOPP (**a**) and ALD-alumina-coated BOPP (**b**).

## Data Availability

The data used to support the findings of this study are already incorporated in the results section (Section 3).

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
