# Peer review of "Exploration of Breakdown Strength Decrease and Mitigation of Ultrathin Polypropylene"

_polymers, 2023, doi:10.3390/polym15102257_

Round 1
Reviewer 1 Report
Ref_comments to the paper titled as “Exploration of Breakdown Strength Decrease and Mitigation of Ultrathin Polypropylene” written by the authors: Daniel Q. Tan1, Yichen Liu, Xiaotian Lin, Enling Huang, Xi Lin, Xudong Wu, Jintao Lin, Ronghai Luo, Tianxiang Wang.
Despite of a large number of works in the polymer materials investigations, including some damage phenomena of the polymers, the estimation of the breakdown strength for the materials with little thickness is very important for the electronics area. From this point of view the current article is actual and modern.
For the first, the authors have made the literature search and have analyzed 17 references. This analysis is really important to support the study of the biaxially oriented polypropylene film (BOPP), used for the capacitors industry. But, it is not enough for this topic. Please add 5-7 manuscripts written and published during last 3 years. It can improve your analysis.
Abstract. Please correct the power density dimension explanation. You have written: “…The breakdown strength drops rapidly and hardly ensures
the capacitor reaches an energy density of 2 J/cm3…” Maybe you have used volumetric energy density?
Experimental section. Please indicate what atomic mass of your polypropylene materials? It should be influence dramatically on the damage level.
Results and Discussion section. Good illustrated and discussed part. The data of damage of the films under the application of the voltage; the SEM-images analysis; heat treatment are presented. In this concern I would like the authors about the UV treatment. Have you’re the data about the breakdown strength of your polypropylene film under the using UV irradiation, maybe at the wavelength of 125, 175 or 190 nm? It is also important when the capacitors can be used at the specific conditions.
Moreover, would you please to explain how your data from Fig.6 (Effect of in-situ heat treatment (115, 150 ºC for 30 minutes) on XRD patterns of BOPP) can be supported via modulated scanning calorimetric (MSC) method?
Conclusion part should be extended.
Please answer the question mentioned above.
As for my local opinion, this paper can be published after minor corrections.
Reviewer 2 Report
The paper presents the study about breakdown strength decrease and mitigation of ultrathin polypropylene. Authors say, that polypropylene film is main dielectric in capacitor, but applications such as power electronic devices require miniaturized capacitors and thinner dielectric films. Presented work studies the breakdown voltage of the film.
Dear author, thank you very much for interesting paper about electrical strength of polypropylene. I put some comments and questions.
Comments:
1. The introduction is well organized. I did not find information about mechanisms of electrical breakdown of dielectric, what is important if we consider that the main topic of the paper is related to breakdown strength. Please complete.
2. Fig.1.a. – it is typical situation, that electrical strength [V/m] decreases with the increase of thickness. It is also typical that if the thickness will be very small, the strength is smaller, because of some contaminants, which may occur inside the dielectric.
3. Fig.3. – please describe in details axis X in the figure – name, units, etc.
4. If authors present alumina coating on the strength, shape and geometry of used electrodes in necessary. Maybe, the coating help to receive more constant electric field distribution [V/m], what is very popular technology used in case of high voltage bushing insulation. Please complete.
